# Effect of CaCO_3_ Nanoparticles on the Mechanical and Photo-Degradation Properties of LDPE

**DOI:** 10.3390/molecules24010126

**Published:** 2018-12-31

**Authors:** Paula A. Zapata, Humberto Palza, Boris Díaz, Andrea Armijo, Francesca Sepúlveda, J. Andrés Ortiz, Maria Paz Ramírez, Claudio Oyarzún

**Affiliations:** 1Grupo Polímeros, Facultad de Química y Biología, Departamento de Ciencias del Ambiente, Universidad de Santiago de Chile, USACH, Santiago 8320000, Chile; borisdiazn@gmail.com (B.D.); andrea.armijot@gmail.com (A.A.); francesca.sepulveda@usach.cl (F.S.); jonathan.ortizn@usach.cl (J.A.O.); maria.ramirezn@usach.cl (M.P.R.); claudio.oyarzun@usach.cl (C.O.); 2Departamento de Ingeniería Química y Biotecnología, Facultad de Ciencias Físicas y Matemáticas, Universidad de Chile, Beauchef 851, Santiago 8320198, Chile; hpalza@ing.uchile.cl

**Keywords:** CaCO_3_ nanoparticles, polyethylene nanocomposites, photoaged polyethylene

## Abstract

CaCO_3_ nanoparticles of around 60 nm were obtained by a co-precipitation method and used as filler to prepare low-density polyethylene (LDPE) composites by melt blending. The nanoparticles were also organically modified with oleic acid (O-CaCO_3_) in order to improve their interaction with the LDPE matrix. By adding 3 and 5 wt% of nanofillers, the mechanical properties under tensile conditions of the polymer matrix improved around 29%. The pure LDPE sample and the nanocomposites with 5 wt% CaCO_3_ were photoaged by ultraviolet (UV) irradiation during 35 days and the carbonyl index (CI), degree of crystallinity (χ_c_), and Young’s modulus were measured at different times. After photoaging, the LDPE/CaCO_3_ nanocomposites increased the percent crystallinity (χ_c_), the CI, and Young’s modulus as compared to the pure polymer. Moreover, the viscosity of the photoaged nanocomposite was lower than that of photoaged pure LDPE, while scanning electron microscopy (SEM) analysis showed that after photoaging the nanocomposites presented cavities around the nanoparticles. These difference showed that the presence of CaCO_3_ nanoparticles accelerate the photo-degradation of the polymer matrix. Our results show that the addition of CaCO_3_ nanoparticles into an LDPE polymer matrix allows future developments of more sustainable polyethylene materials that could be applied as films in agriculture. These LDPE-CaCO_3_ nanocomposites open the opportunity to improve the low degradation of the LDPE without sacrificing the polymer’s behavior, allowing future development of novel eco-friendly polymers.

## 1. Introduction

Inorganic fillers are incorporated into a polyolefin to form composites with enhanced mechanical, thermal, and barrier properties compared to the polymer matrix [1]. Nanocomposites are a class of filled polymers in which nanometric inorganic fillers are incorporated into the polymer matrix with property enhancements at much lower concentrations than those of microfillers. Calcium carbonate is one of the most commonly used inorganic fillers in thermoplastic polymers, such as poly(vinyl chloride) and polypropylene, to improve their mechanical properties. CaCO_3_ nanoparticles, in particular, have been incorporated into a polyethylene (PE) matrix by the melting process, increasing Young’s modulus with the filler concentration and decreasing both the upper yield point and elongation at break compared to pure PE [2,3,4,5]. Although CaCO_3_ is a well-known filler in polymer composites for mechanical reinforcement, at the nanometric scale it can add other functionalities to the polymer matrix, such as barrier properties and antimicrobial behavior [6,7].

Recently, the effect of the incorporation of nano-particulate calcium carbonate hollow spheres (3, 10 and 25 wt%) in high-density polyethylene (HDPE) by extrusion was studied. They found a crystallinity decrease with increasing filler content. There was found a typical increase of Young’s modulus (E) (ca. 17%) with increasing concentration of hollow spheres of CaCO_3_ filler due to the rigidity of the filler particles and the strong interaction of the filler with the polymer matrix, and it was companied by the corresponding decrease of the upper yield point and elongation at break [2].

One of the major drawbacks of polymer nanocomposites is the high agglomeration of the fillers, which can be reduced by surface modifications. For instance, Lazzeri et al. [8] studied the influence of the organic surface modification in CaCO_3_ nanoparticles (70 nm) by stearic acid (SA) treatment on the mechanical properties of HDPE composites. Incorporation of 10 vol% of CaCO_3_ to HDPE increased a rise in yield stress in all composites, but the yield stress decreases with increasing SA content. The author explained this behavior by stating that the addition of SA to the surface of the particles should reduce the stress transfer ability of the interface and even its thickness, leading to a softer interface. The influence of nano-CaCO_3_ and its surface modification have also been studied in polypropylene (PP) composites prepared by the melting process. In particular, nano-CaCO_3_ (diameter ca. 44 nm) was modified with stearic acid [1,9,10] and palmitic acid [11], with the addition of modified CaCO_3_ increasing tensile strength, Young’s modulus, and melting point. Another route to improve the dispersion of nano-CaCO_3_ in PP matrices is by the addition of a small amount of a non-ionic modifier during melt extrusion. In this case Young’s modulus increased slightly with amount of CaCO_3_ load, while the yield strength of PP decreased [12].

On the other hand, to improve the physicochemical properties of the polymer some researchers have treated the surface polymer using thin-layer technology, including oxygen and nitrogen plasma discharge, deposition of functional coatings (i.e., diamond-like carbon (DLC)) among others. For example, low-density polyethylene (LDPE) increased its surface hardness 7 times after layer deposition by DLC coating. Those techniques allowed giving desirable surface properties to the polymer [13].

Despite the relevance for society of the degradation properties of plastic materials, the environmental stability of polymer/calcium carbonate nanocomposites has been barely reported. The effect of nano-CaCO_3_ on the natural photo-aging degradation of PP was studied outdoors during 88 days [14]. The degradation polymer was studied by Fourier transform infrared (FTIR) spectroscopy and pyrolysis gas chromatography-mass spectroscopy (PGC-eMS). The PP/CaCO_3_ nanocomposites showed higher photo-degradation than neat PP. The authors explained this behavior as due to the functional groups on the surface of nanoparticles catalyzing the photo-oxidation reaction of PP. There are adsorbed hydroxyl groups on the surface, which is active in photo-chemical reactions. Morreale et al. [15] studied the accelerated weathering behavior of PP/CaCO_3_ micro- and nanocomposites, showing that the nanosized filler may lead to a faster photo-oxidation rate than that of pure polypropylene. In particular, nanosized calcium carbonate caused faster photodegradation rates than microsized calcium carbonate.

Achieving the bidodegradation of commercial commodity plastics is an enormous environmental challenge due to the increased social demand for higher sustainability processes. The addition of additive/filler to accelerate the photodegradation of these polymers can be associated with an early decrease in the mechanical property even during use. Therefore, nanoparticles able to accelerate the photodegradation together with improving the mechanical behavior can compensate for the latter issue.

Considering what was mentioned above, the present work studies the effect of adding pure and organic-modified CaCO_3_ nanoparticles into a non-polar LDPE matrix. The effect of different amounts of pure CaCO_3_ nanoparticles and oleic acid-modified-CaCO_3_ on the thermal and mechanical properties of polyethylene were studied. The effect of CaCO_3_ nanoparticles on the photoaging process of LDPE was further investigated.

## 2. Experiment

### 2.1. Materials

Polyethylene was purchased from Aldrich, density: 0.925 g cm^−3^, melt index: 25 g/10 min (190 °C/2.16 kg); impact strength: 45.4 J/m (Izod, ASTM D 256, −50 °C). CaCO_3_ nanoparticles were synthesized by a precipitation method [16]. Briefly, the reagents used were sodium carbonate, Na_2_CO_3_ (Merck, Darmstadt, Germany, 99.9%), calcium nitrate, Ca(NO_3_)_2_ (Aldrich, Darmstadt, Germany, 99%), sodium nitrate NaNO_3_ (Aldrich, 99%), sodium hydroxide, NaOH pellets (Mallinckrodt Chemicals., Dublin, Ireland, ≥98%), and distilled water. Oleic acid (Aldrich, reagent grade, 98%) was used for the modification of the CaCO_3_ nanoparticles.

### 2.2. CaCO_3_ Nanoparticle Synthesis

The CaCO_3_ nanoparticles were obtained by a method reported by Babou-Kammoe et al. [16]. First, sodium carbonate (Na_2_CO_3_) (0.042 g) was dissolved in deionized water (80 mL) with sodium hydroxide (NaOH) (1.25 g) and sodium nitrate NaNO_3_ (0.612 g). In a second step, calcium nitrate (Ca(NO_3_)_2_) (0.944 g) was dissolved in deionized water (80 mL) and the resultant mixture formed a precipitate. The calcium nitrate solution was added dropwise to the sodium carbonate solution with continuous stirring during 4 h at 25 °C. The resultant mixture formed a precipitate which was separated from the water by filtering off. The nanoparticles were dried at 60 °C during 24 h and were characterized.

### 2.3. Organic Modification of CaCO_3_ Nanoparticles (O-CaCO_3_)

The nanoparticles were modified with oleic acid [17]. 1-Hexane (100 mL) and oleic acid (200 µL) were mixed with stirring. Then 1 g of CaCO_3_ nanoparticles was added to the solution at 60 °C with vigorous stirring during 5 h. The nanoparticles were then filtered, washed with ethanol, and vacuum-dried at 100 °C during 24 h [18].

### 2.4. Low-Density Polyethylene (LDPE)/CaCO_3_ and LDPE/O-CaCO_3_ Nanocomposite

The nanocomposites were prepared using a Brabender Plasti-Corder (Duisburg, Germany) internal mixer at 150 °C and a speed of 110 rpm, during 10 min. The nanocomposites with 3, 5, and 8 wt% of CaCO_3_ nanoparticles were obtained by mixing predetermined amounts of the CaCO_3_ as filler and neat LDPE under a nitrogen atmosphere. The samples were press-molded at 190 °C at a pressure of 50 bar during 3 min and cooled under pressure by flushing the press with cold water.

### 2.5. Nanoparticles and Composite Characterization

The morphology of the CaCO_3_ was analyzed by transmission electron microscopy (TEM) (JEOL ARM 200 F, Boston, MA, USA) operating at 20 kV. Samples for TEM measurements were prepared by placing a drop of CaCO_3_ nanoparticles on a carbon-coated standard copper grid (400 mesh).

The X-ray diffraction (XRD) patterns of the CaCO_3_ nanoparticles were studied on a Siemens D5000 diffractometer (Berlin, Germany), using Ni-filtered Cu Kα radiation (λ = 0.154 nm). The diffraction patterns were recorded in the 2θ = 5–80° range.

FTIR measurements of CaCO_3_ and modified nanoparticles (O-CaCO_3_) were performed in a Bruker Vector 22 FTIR spectrometer (Karlsruhe, Germany). The infrared (IR) spectra were collected in the 4000 to 500 cm^−1^ range, with a resolution of 4 cm^−1^ at room temperature.

The tensile properties of the neat polyethylene (neat LDPE) and composites (LDPE/CaCO_3_ and LDPE/O-CaCO_3_) were determined on an HP model D-500 dynamometer (Buenos Aires, Argentina). The materials were molded for 3 min in a hydraulic press, HP Industrial Instruments, at a pressure of 50 bar and a temperature of 170 °C, and then cooled under pressure with water circulation. Films around 0.05 mm thickness were obtained. Dumbbell-shaped samples with an effective length of 30 mm and a width of 5 mm were cut from the compression-molded sheets. The samples were tested at a rate of 50 mm/min at 20 °C. Each set of measurements was repeated at least four times.

#### 2.5.1. Photo-Exposure

##### Photoaging

Polymer films of 0.02 mm of thickness and the dimensions and 4 cm were irradiated using a Microscal Light exposure unit and Suntest/Atlas XLS 2200 W (Linsengericht, Germany) using a solar standard filter (borosilicate), which provides 550 W m^−2^ (Irradiance acc. ISO 4892/DIN 53387) in the 300–800 nm wavelength region. The temperature was kept constant at 45 °C during the testing. Exposed samples of 1 cm × 1 cm were periodically taken out and characterized. The irradiation side of the sample was alternated every 3 days. At different aging times the oxidation rates were determined on an FTIR spectrometer using the standard carbonyl index method. FTIR spectra were obtained on a Perkin Elmer BX-FTIR (Waltham, MA, USA). The polymer degradation was determined using the carbonyl index (CI) as the ratio of the optical density of the ketone carbonyl absorptions bands at 1715 cm^−1^ and the optical density corresponding to CH_2_ scissoring peak at 1465 cm^−1^ [19].

Differential scanning calorimetry (DSC) was studied on a METTLER DSC823 (Columbus, OH, USA) The melting temperature and enthalpy of fusion of the neat and nanocomposite samples were determined before and after photoaging. The measurements were made at a heating rate of 10 °C·min^−1^ in an inert atmosphere. The samples were heated from 25 °C to 180 °C and then cooled to 25 °C at the same rate. Percent crystallinity (χ_c_) was determined using Equation (1):(1)$$\chi c=\;\frac{\Delta H_{l}}{\left(1\;-\;\Phi \right)\Delta H_{0}}\times 100$$
where $\Delta H_{l}$ is the melting enthalpy (J g^−1^) of the polymer nanocomposite, $\Delta H_{0}$ is the enthalpy corresponding to the melting of a 100% crystalline sample (289 J g^−1^) [20], and Φ is the weight fraction of the filler in the nanocomposit. The standard deviation of the T_c_ and T_m_ measurements was ca. ±2 °C.

Thermogravimetric analysis (TGA) experiments were performed on a Netzsch TG 209 F1 Libra Instrument (Selb, Germany). The films were heated from 25 °C to 600 °C at a rate of 10 °C·min^−1^ and the nitrogen flow was kept constant at 60 mL·min^−1^. The TGA analysis also verified the content of CaCO_3_ in the LDPE/CaCO_3_ nanocomposites. The LDPE/CaCO_3_ nanocomposites with 5 wt% showed a 4.65 wt% of the CaCO_3_ nanoparticle content after the melting process.

Viscosimetric analysis before and after irradiations was carried out in o-dichlorobenzene at 135 °C in a Viscosimatic-Sofica viscometer (Santiago, Chile)

The surface morphology of the polymers before and after photoaging was characterized by scanning electron microscopy (SEM) using a Philips XL30 model instrument (Billerica, MA, USA).

## 3. Results and Discussion

### 3.1. Nanoparticle Characterization

The morphology of the nanoparticles was studied by TEM as shown in Figure 1. The nanoparticles synthesized by the coprecipitation method have an average diameter of ca. 60 nm and irregular morphology. The yield of this method was ca. 75%. The crystalline phases of the CaCO_3_ nanoparticles were studied by XRD (Figure 2). Nanoparticles have two characteristic phases as concluded by analyzing the Bragg reflections: calcite, associated with peaks at 29° and 32° from the (104) and (006) crystal planes, respectively; and aragonite, with peaks at 27°, 30°, and 45° from the (111), (021), and (221) planes, respectively [21].

The FTIR spectrum of calcined samples (CaCO_3_) (Figure 3b), shows the presence of calcium carbonate (CaCO_3_) bands at 715, 880, 1490, 1804, 2530, 2900, and 2998 cm^−1^. The band at 1490 cm^−1^ correspond to essentially asymmetric and symmetric lengthening of the O–C–O bond. Absorption bands centered at 715, 880 and 1490 cm^−1^ are characteristics of the calcite phase of CaCO_3_ [22]. The method used for the organic modification of nanoparticles was based on that reported by Li and Zhu [17] and it was verified by FTIR spectra, where the peaks corresponding to the alkyl chain (CH_2_) of oleic acid appear at 2920 cm^−1^ and 2855 cm^−1^ (Figure 3). Moreover, a small spectral line at 1710 cm^−1^ corresponding to the stretching of the carbonyl group of oleic acid indicates that the carboxylic acid group of oleic acid, −COOH, reacted with surface hydroxyl groups from the starch nanoparticles [23]. Other peaks at 1590 cm^−1^ due to carboxylate groups, and the peaks at 1550 cm^−1^ and 1430 cm^−1^ that indicate the presence of COO−, are overlapped with the characteristic band at 1490 cm^−1^ of the O–C–O bond and calcite vibrations.

### 3.2. Composite Characterization

#### 3.2.1. Thermal Properties

The crystallization temperature (T_c_), melting temperature (T_m_), and degree of cristallinity (χ_c_) were analyzed by DSC and the thermal stability obtained by TGA of the neat LDPE and LDPE/CaCO_3_ nanocomposites are shown in Table 1. The crystallization temperature, melting temperature, and degree of crystallinity (χ_c_) did not change with the incorporation of the nanoparticles, meaning that the presence of these nanoparticles did not affect the crystallization process of the polymer matrix. Similar results have been reported by other authors when different nanoparticles like ZnO, clay, silica, silver, and TiO_2_ were incorporated into LDPE. This behavior may be correlated to the minimal volume fraction of the nanoparticles incorporated into the composite [24,25,26]. Also, the similar thermal properties of the matrix and composites would suggest analogous processing conditions as that of LDPE at a hypothetical industrial-scale production of these nanocomposites.

In the initial degradation step of the decomposition temperature, at 2% weight loss (T_2_), the nanocomposites (LDPE/CaCO_3_) were slightly more stable than LDPE nanocomposites at ca. 5%. For 10% weight loss (T_10_), for 50% weight loss (T_50_), and the temperature for the maximum rate of weight loss (T_max_) did not change with the nanoparticle incorporation compared to the pure neat LDPE under inert conditions. It is well known that the incorporation of different kinds of nanofillers into a polymer can act as a superior insulator and mass transport barrier for the volatile products generated during decomposition, increasing the thermal degradation temperatures. However, these processes are relevant for high aspect ratio nanoparticles such as layered clays. Spherical-like particles with low aspect ratio should not trigger these mechanisms and only an adsorption process can explain changes in the degradation, as reported by our group on spherical silica nanoparticles [27]. In our case, the spherical-like CaCO_3_ nanoparticles were not able to disrupt the diffusion nor adsorb volatile compounds and, therefore, no changes were observed in TGA analysis.

#### 3.2.2. Mechanical Properties

The mechanical properties of the neat LDPE and the LDPE/CaCO_3_ nanocomposites are displayed in Table 2 and Figure 4. An increase of Young’s modulus results from adding the CaCO_3_ nanoparticles in comparison with the neat LDPE. This performance was more pronounced with 5 wt% of nanoparticles for the LDPE/O-CaCO_3_ nanocomposites, as Young’s modulus increased ca. 29% compared to neat LDPE. Morreale et al. [3] found that 10 wt% of CaCO_3_ fillers (50–100 nm) improved Young’s modulus just ca. 20% compared to neat LDPE, due to the presence of the nanoparticle agglomeration. The increase in the modulus in our case must be caused by the strong interaction between the polymer and the nanoparticles, improving the dispersion of the particles [1]. Similar results were found by Lapcík et al. [2], who stated that due to the rigidity of the filler particles and the interaction of the filler with the polymer matrix, a reinforcement improvement can be obtained. The yield stress remained unaffected with the addition of CaCO_3_ to neat polyethylene. A similar behavior was found for nanocomposites based on high-density polyethylene with CaCO_3_ nanoparticles (ca. 60 nm) [9].

On the other hand, the deformation at break decreased when 5 wt% of the CaCO_3_ nanoparticles were incorporated, probably due to many defects in the polymer matrix leading to ductility reduction [10]. The decrease of the deformation at break of LDPE/O-CaCO_3_ (5 wt%) was slightly lower, and this may be due to the modifier improving the interaction between nanoparticles and LDPE [1].

### 3.3. Photoaging Analysis

#### 3.3.1. Thermal and Mechanical Properties

Crystallization temperature (T_c_), melting temperature (T_m_), degree of cristallinity (χ_c_), thermal stability analysis, and viscosity of the neat LPE and LDPE/CaCO_3_ nanocomposites after photoaging are displayed in Table 3. After photoaging, the χ_c_ for nanocomposites increased slightly compared to pure LDPE and LLDPE/CaCO_3_ before irradiation. This behavior has been attributed to recrystallization due to LDPE scission of end chains producing mobile small chain fragments able to undergo reorganization and recrystallization [14,28]. This scission is confirmed also finding that after photoaging the viscosity decreased due to the formation of low molecular weight compounds during aging (Table 3). It should be noted that this behavior is slightly greater for nanocomposites than for neat LDPE. These results show that the incorporation of nanoparticles into the polymer accelerates its degradation. After photoaging, both decomposition temperatures, (T_10_) and T_max_, did not change. In previous work using Ca and Fe stereates as PE degradant, the authors found a slight decrease in T_10_, and they explained this behavior as due to the prooxidative nature of stereate during the photoaging process [29].

The mechanical properties were evaluated after 10 days of photoaging as displayed in Table 4. The polymers were difficult to break into pieces by hand after 10 days of irradiation, confirming the strong degradation. Young’s modulus for the LDPE/CaCO_3_ increased after photoaging compared to photoaged neat PE. Young’s modulus increased after photo-oxidation mainly due to a significant embrittlement of the material and recrystallization phenomena caused by scission reactions [15]. These results further confirmed that nanoparticles accelerated the degradation of the polymer.

#### 3.3.2. Carbonyl Index

The degradation, calculated by FTIR, and the carbonyl index (CI) after 35 days of irradiation are displayed in Figure 5. The carbonyl index was measured for neat LDPE and the LDPE/CaCO_3_ sample with 5 wt% of nanoparticles. The LDPE/CaCO_3_ nanocomposite with 5 wt% had a higher carbonyl index than LDPE, showing an influence in the degradation of the polymers with nanoparticle incorporation. The IR spectra of the photoaged polymer (LDPE/CaCO_3_) has a strong peak at 1720 cm^−1^, which is related to the C=O stretching vibration of the carbonyl group (Figure 6) [30]. The second band around 3400 cm^−1^ is related to the hydroxyl group, which indicates the generation of hydroperoxides and hydroxyl species. Furthermore, the intensity of the carbonyl and hydroxyl bands grew with increasing exposure time [14]. The intensity peaks at 2930, 2850, 1470, and 720 cm^−1^, corresponding to the alkyl chain, decreased slightly. Carboxylic acid salts can be formed by the reaction between the carboxylic acids coming from photoaged PE and the basic fillers. Li et al. [14] explained that PP/CaCO_3_ shows a higher degradation rate than neat PP, due to functional groups on the surface of the nanoparticles catalyzing the photooxidant ion reaction of PP. There are absorbed hydroxyl groups on the surface of the nanofillers, which are active in photo-chemical reactions. Therefore, the hydrophilic surface of the nanoparticles is responsible for the increased polymer degradation. Further degradation in the abiotic environment is through the Norrish type I and II mechanism, giving rise to esters and ketones [31].

SEM images of LDPE and LDPE/CaCO_3_ with 5 wt% of CaCO_3_ before and after photoaging are shown in Figure 7. Before irradiation, LDPE and LDPE/CaCO_3_ images exhibit a smooth and homogeneous surface morphology (Figure 7a,c). After irradiation, the morphology is changed, the LDPE and LDPE/CaCO_3_ nanocomposites presented some cavities, with nanoparticles producing larger ones (Figure 7b,d). After irradiation, the nanocomposites undergo greater acceleration of photodegradation than neat LDPE, confirming the results shown above by CI, mechanical properties, and viscosity.

## 4. Conclusions

The co-precipitation method was used to produce CaCO_3_ (60 nm), which were then modified organically with oleic acid (O-CaCO_3_). Young’s modulus increased ca. 29% for LDPE/O-CaCO_3_ compared to the neat LDPE.

Regarding polymer photoaging, the degree of crystallinity (χ_c_) increased with photoaging, and this effect was higher for LDPE/CaCO_3_ (ca. 19%) nanocomposites than for neat LDPE (ca. 8%), attributed to recrystallization of the polymer. The viscosity of LDPE decreased by ca. 59% after photoaging and around 72% for LDPE/CaCO_3_, as indicated by the decreased molecular weight of the polymer due to chain scissions, and the pronounced effect of the nanoparticles in the polymer degradation. Young’s modulus increased ca. 16% for LDPE/O-CaCO_3_ after photoaging because the nanoparticles accelerate the polymer’s degradation. The degradation of the films obtained was confirmed by the carbonyl index, where carbonyl bands appear more intense. LDPE/CaCO_3_ with 5 wt% had a high carbonyl index, showing an influence in the degradation of the polymers with the incorporation of nanoparticles.

## Figures and Tables

**Figure 1 molecules-24-00126-f001:**
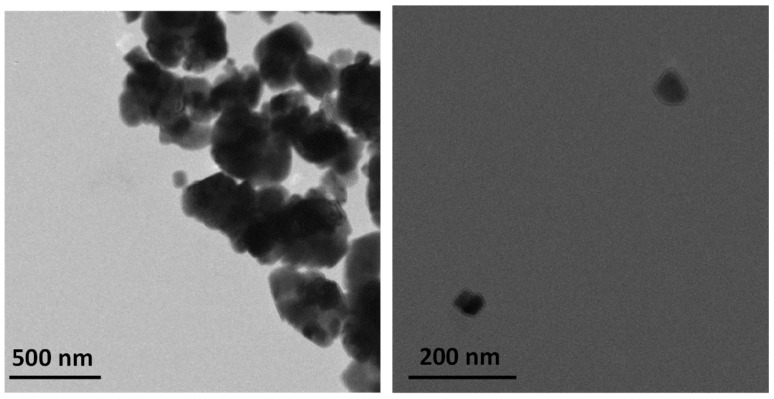
Transmission electron microscopy (TEM) image of the CaCO_3_ nanoparticles.

**Figure 2 molecules-24-00126-f002:**
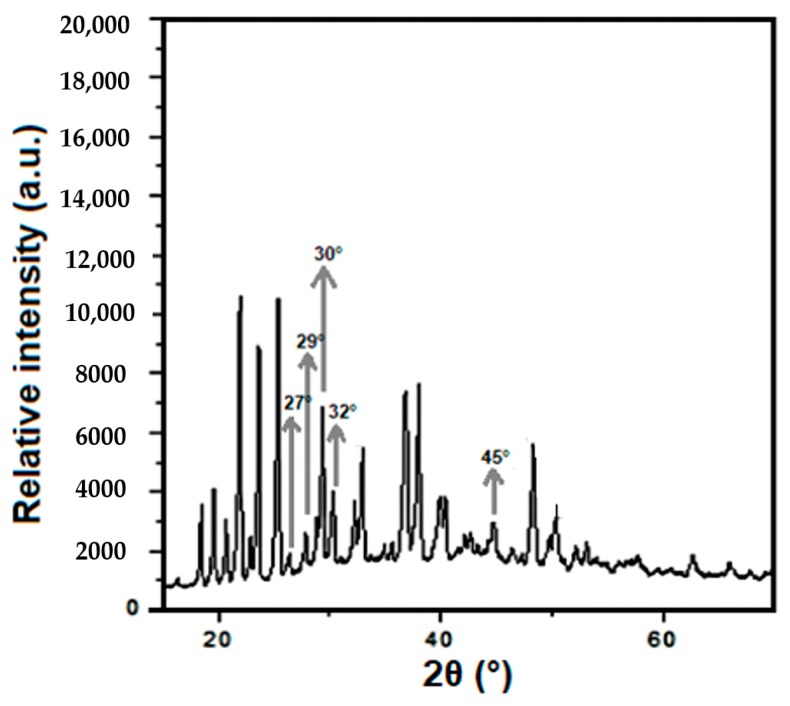
X-ray diffraction (XRD) spectra of CaCO_3_ nanoparticles.

**Figure 3 molecules-24-00126-f003:**
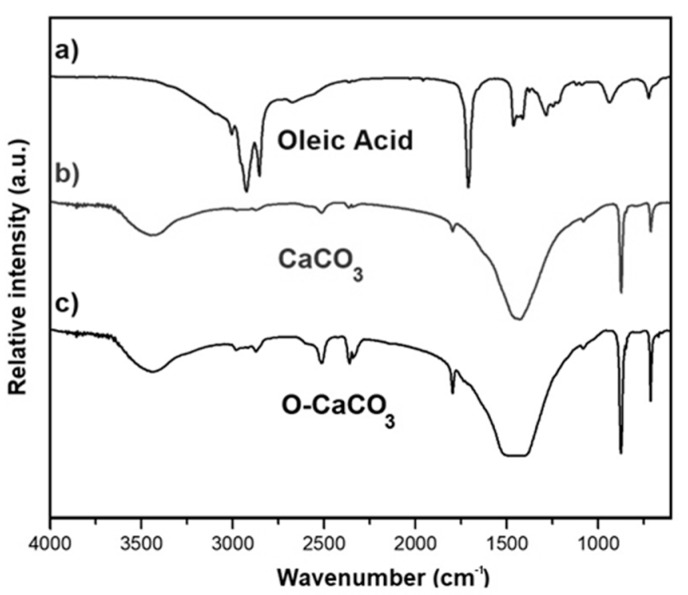
Fourier transform infrared (FTIR) spectra of (**a**) Oleic acid, (**b**) CaCO_3_ nanoparticles, and (**c**) nanoparticles modified with oleic acid (O-CaCO_3_).

**Figure 4 molecules-24-00126-f004:**
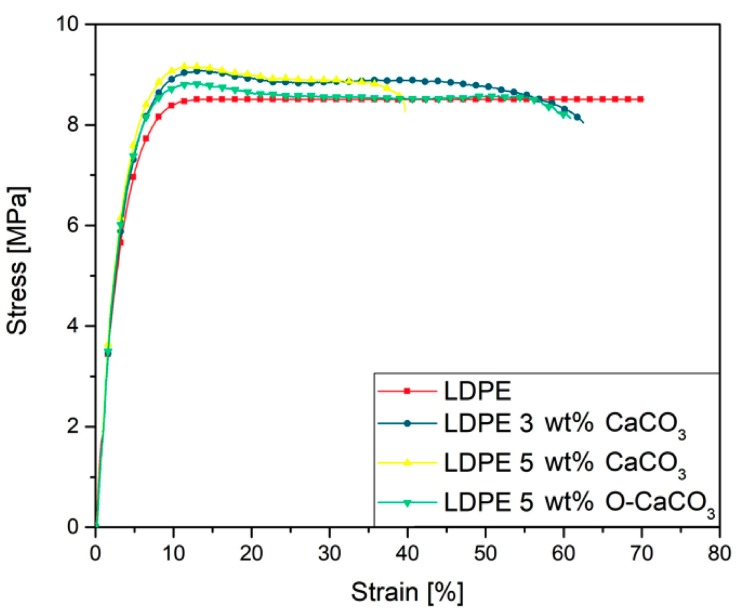
Stress-strain curves for neat LDPE, LDPE/CaCO_3_ with 3 and 5 wt%, and LDPE/O-CaCO_3_ with 5 wt% nanocomposites.

**Figure 5 molecules-24-00126-f005:**
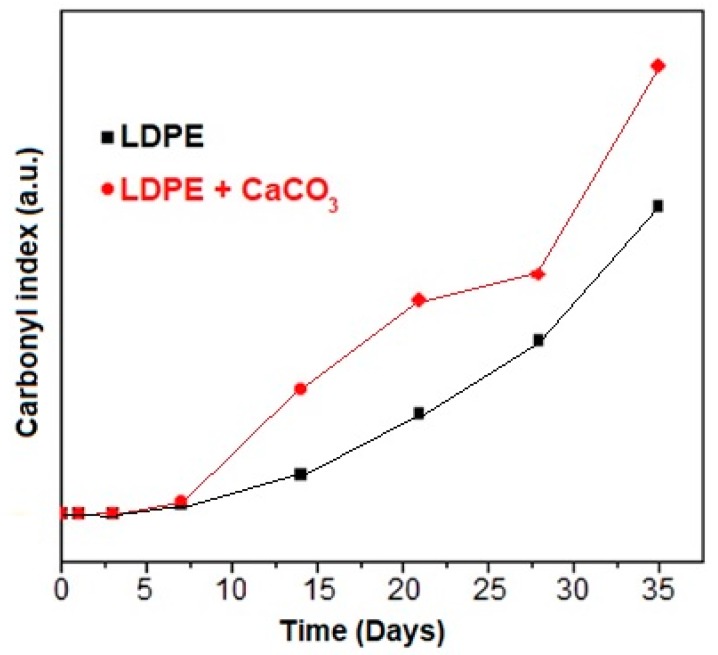
Carbonyl index (CI) of neat LDPE and LDPE/CaCO_3_ at different irradiation times.

**Figure 6 molecules-24-00126-f006:**
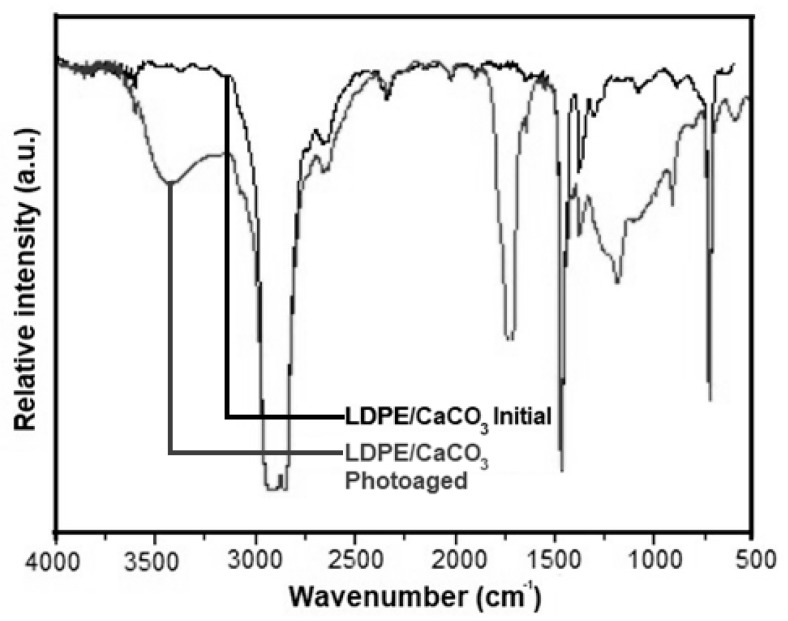
Infrared (IR) spectra of initial LDPE/CaCO_3_ nanocomposites and LDPE/CaCO_3_ after photoaging for 35 days.

**Figure 7 molecules-24-00126-f007:**
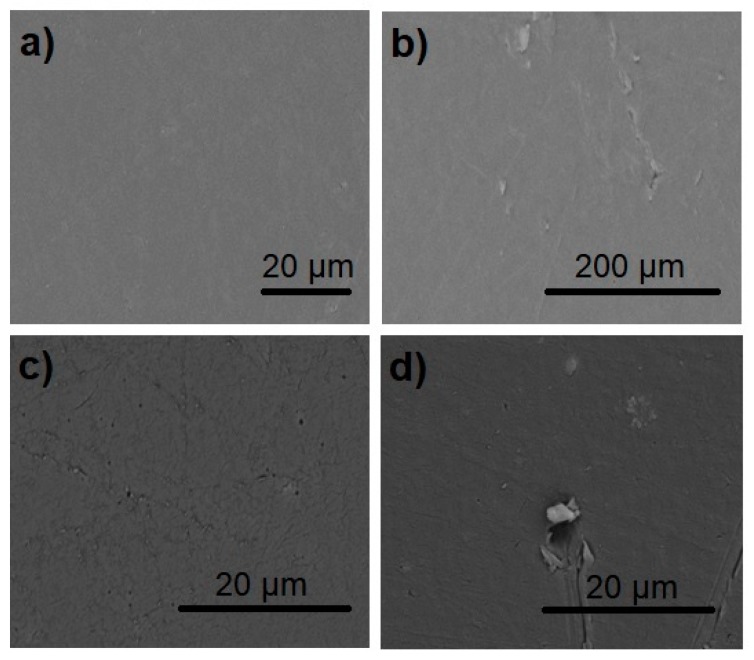
Scanning electron microscopy (SEM) images of initial and photoaged PE and PE/SNp during 35 days of photoaging: (**a**) PE initial; (**b**) PE aged; (**c**) PE/CaCO_3_ initial; and (**d**) PE/CaCO_3_ aged.

**Table 1 molecules-24-00126-t001:** Thermal properties of polyethylene (PE)/CaCO_3_ nanocomposites before photoaging.

Process	CaCO_3_ (wt%)	η (dL/g)	T_c_ (°C)	T_m_ (°C)	χ_c_ (%)	T_2_ (°C)	T_10_ (°C)	T_50_ (°C)	T_max_ (°C)
Neat low-density polyethylene (LDPE)	N/A	0.44	100	112	37	385	421	459	464
LDPE/CaCO_3_	5	0.46	100	111	37	396	421	456	460
LDPE/O-CaCO_3_	5	0.46	101	111	35	404	420	456	462

T_c_: Crystallization temperature, η: Viscosity, T_m_ = melting temperature; χ_c_ = percent crystallinity, T_2_ = decomposition temperature at 2% weight loss; T_10_ = decomposition temperature at 10% weight loss; T_50_ = decomposition temperature at 50% weight loss, T_max_ = temperature for the maximum rate of weight loss (T_max_); O-CaCO_3_ = modified nanoparticles. The standard deviation of the viscosity measurements is ±0.03 dLg^−1^. The standard deviation of the T_m_ and T_c_ measurements are ca. ±2 °C. The thermogravimetric analysis (TGA) has a standard deviation of ca. ±2 °C.

**Table 2 molecules-24-00126-t002:** Mechanical properties of LDPE and LDPE/CaCO_3_ nanocomposites.

Process	CaCO_3_ Content (wt%)	E (MPa)	σy (MPa)	ε _Break_ (%)
Neat LDPE	0	202 ± 7	8.5 ± 0.03	70.3 ± 8
LDPE/CaCO_3_	3	230 ± 7	9.1 ± 0.15	62.5 ± 10
	5	250 ± 4	9.2 ± 0.09	39.8 ± 3
LDPE/O-CaCO_3_	5	260 ± 10	8.8 ± 0.14	61.1 ± 1

E = Young’s modulus; σy = yield stress; ε _Break_ = deformation at break.

**Table 3 molecules-24-00126-t003:** Thermal properties of PE/CaCO_3_ nanocomposites after photoaging.

				Photoaging			
Nanoparticles	CaCO_3_ (wt%)	η (dL/g)	T_c_ (°C)	T_m_ (°C)	χ_c_	T_10_ (°C)	T_max_ (°C)
Neat LDPE	N/A	0.18	105	107	40	423	467
LDPE/CaCO_3_	5	0.13	106	108	44	426	460

T_c_: crystallization temperature, T_m_ = melting temperature; χ_c_ = percent crystallinity. T_10_ = decomposition temperature at 10% weight loss; T_max_ = temperature for the maximum rate of weight loss (T_max_). Photoaging during 10 days.

**Table 4 molecules-24-00126-t004:** Mechanical properties of LDPE and LDPE/CaCO_3_ nanocomposites after irradiation during 10 days.

		Photoaging		
Nanoparticles	CaCO_3_ (wt%)	E (MPa)	σy (MPa)	ε _Break_ (%)
Neat LDPE	N/A	214 ± 14	7 ± 0.3	20 ± 4
LDPE/CaCO_3_	5	301 ± 15	6 ± 0.3	8 ± 3

E = Young’s modulus; σy = yield stress; ε _Break_ = deformation at break. The photoaging during 10 days.

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
