# Peer review of "Effect of CaCO3 Nanoparticles on the Mechanical and Photo-Degradation Properties of LDPE"

_molecules, 2018, doi:10.3390/molecules24010126_

Round 1
Reviewer 1 Report
General comment The title of this paper is misleading. It suggests that addition of CaCO3-nanoparticles improves mechanical properties and weakens the effect of photo degradation. Instead, the only property that improves is the Young’s modulus. Which is no new insight. The question you aim at is important: If you fill a polymer with nanoparticles, what do I have to do to compensate for the accelerated photo-degradation. The paper does not give an answer to this. The text must be checked by an English native speaker. You used TGA, but you do not mention in the paper how you verified the content of CaCO3 in your mixtures. Detail comments Chapter 2.1. This is the worst material’s characterisation I’ve ever encountered. Just the density? How about the product name, molecular weight, melt flow index, zero shear viscosity? In particular the melt viscosity at the compounding temperature would be important. In Tables 2 and 4, there are many typos in the header.
Author Response
December 10, 2018 , Chile
Professor
Editor-in-Chief
Molecules
Dear Editor
First of all, we would like to thank the reviewers for their revision, which was very helpful to improve the manuscript.
Please find enclosed with this note our responses to the reviewers regarding their comments to our manuscript entitled: “Effect of CaCO3 nanoparticles on the mechanical and photo-degradation of LDPE ”
Reviewer 1
General comment The title of this paper is misleading.
1. It suggests that addition of CaCO3-nanoparticles improves mechanical properties and weakens the effect of photo degradation. Instead, the only property that improves is the Young’s modulus. Which is no new insight.
ANSWER: Thanks to Reviewer 1 for this comment. The title of the paper was modified to avoid misleading. The new title is: “Effect of CaCO3 nanoparticles on the mechanical and photo-degradation of LDPE”. On the other hand, our results showed that after photo degradation tests by UV irradiation the nanocomposite presented higher carbonyl index (CI), degree of crystallinity (%χ), and Young’s Modulus than pure polymer. Moreover, the viscosity of the nanocomposite decreased as compared with the photoaged pure LDPE. These changes are due to the scission of the polymer chain meaning that nanoparticles improve the photodegradation of the polymer matrix. The summary was changed stressing this point.
2.The question you aim at is important: If you fill a polymer with nanoparticles, what do I have to do to compensate for the accelerated photo-degradation. The paper does not give an answer to this.
Thanks to Reviewer 1 for this point of view that actually was not considered in our original manuscript, and for that reason was not answered explicitly.
ANSWER:The main question of our original manuscript was if CaCO3 nanoparticles can be considered as multifunctional filler capable of improving simultaneously the mechanical property of the polymer and the photodegradation, both properties considered independently. The point of view of Reviewer 1 is complementary to our original question as the improved mechanical property of the nanocomposite (both in the original sample and during aging) can compensate for the effect of the accelerated photo-degradation.
On page 11, the new paragraph was incorporated:
The bidodegradation of commercial commodity plastics is an enormous environmental problem due to the increased social demand for higher sustainability processes. The addition of additive/filler to increase the photodegradation of these polymers can have the issue associated with an early decrease in the mechanical property even during use. Therefore, nanoparticles able to improve the photodegradation together with improving the mechanical behavior can compensate for the last issue. These LDPE-CaCO3 nanocomposites open the opportunity to improve the low degradation of the LDPE without sacrificing the polymer behavior allowing future development of novel eco-friendly polymers.
3.The text must be checked by an English native speaker.
ANSWER:The new manuscript version was corrected by an experienced English speaker
4.You used TGA, but you do not mention in the paper how you verified the content of CaCO3 in your mixtures.
The new sentence in the experimental part is:
ANSWER: The TGA analysis also verified the content of CaCO3 in the LDPE/CaCO3 nanocomposites. The LDPE/CaCO3 nanocomposites with 5 wt.% showed a 4.65 wt% of the CaCO3 nanoparticle content after the melting process.
5. Detail comments Chapter 2.1. This is the worst material’s characterisation I’ve ever encountered. Just the density? How about the product name, molecular weight, melt flow index, zero shear viscosity? In particular the melt viscosity at the compounding temperature would be important.
ANSWER:You are right about completing the material specification of the LDPE.
The new sentence is:
Polyethylene was purchased from Aldrich, density: 0.925 g.cm-3, melt index: 25 g/10 min (190°C/2.16 kg); impact strength: 45.4 J/m (Izod, ASTM D 256, −50 °C).
The compounding temperature is described in section 2.4, in the next sentence:
The composites were prepared using a Brabender Plasti-Corder internal mixer at 150 ºC and a speed of 110 rpm during 10 minutes.
6.In Tables 2 and 4, there are many typos in the header.
It was corrected
Reviewer 2 Report
The paper submited by Zapata et al. is well-written and presents interesting results however some changes need to be done before taking into consideration publishing it Molecules Journal.
Comments:
Some words stick together, for instance: p.1., l. 30/31, then line 42, etc. Please read the text carefully after exporting into pdf file.
Please correct kv into kV- page 3, l. 102.
XRD analysis - provide the range within the measurement has been performed.
Tensile measurements- following which standard did you perform the analysis.
Please provide the equation, that has been used to calculate the Xc- did you subtract the amount of the nanofiller from the enthalpy in the denominator of the equation?
There are few grammar errors that have crept into the paper, for example: page 4, line 154: the FTIR spctrum...show", etc.
Did you estimate also Tc besides Tm. The analysis on the influence of the addition of nanofiller on the crystallinity of LDPE should be expanded upon the discussion on Tc. Maybe authors can add such data in the Table 1.
In my opinion, not only T10 should be presented. Maybe also the begining of degradation process - T2% and the middle of this process: T50% can be presented. Than one can observe how the addition of nanofillers affect the decomposition process on every step of degradation.
Point 3.2.2-please provide the stress-strain curves for the samples. In my opinion the data presented in table are not enough for the analysis.
There are only 8 from 29 references published from the late 5 years- which means less than 30% - maybe authors can find more actual references that can be cited in the paper.
TRherefore I recommend it for majro revision.
Author Response
December 10, 2018 , Chile
Professor
Editor-in-Chief
Molecules
Dear Editor
First of all, we would like to thank the reviewers for their revision, which was very helpful to improve the manuscript.
Please find enclosed with this note our responses to the reviewers regarding their comments to our manuscript entitled: “Effect of CaCO3 nanoparticles on the mechanical and photo-degradation of LDPE ”
Reviewer 2
1. Some words stick together, for instance: p.1., l. 30/31, then line 42, etc. Please read the text carefully after exporting into pdf file.
Therefore, I recommend it for major revision.
ANSWER: It was corrected
2- Please correct kv into kV- page 3, l. 102.
ANSWER: It was corrected.
3.XRD analysis - provide the range within the measurement has been performed.
ANSWER: The Diffraction patterns were recorded in the 2θ = 5– 80° range.
4. Tensile measurements- following which standard did you perform the analysis.
ANSWER: The Tensile measurements were measurement for neat polyethylene (neat LDPE) without nanoparticle incorporation and for the LDPE/CaCO3 and LDPE/O-CaCO3· nanocomposites. The neat LDPE was the reference polymer in order to study the effect of the nanoparticle incorporation on the mechanical properties. The measurements of the neat LDPE and LDPE/CaCO3 were made under the same conditions.
In the manuscript the reference polymer, neat polyethylene, was called neat LDPE.
5. Please provide the equation, that has been used to calculate the Xc- did you subtract the amount of the nanofiller from the enthalpy in the denominator of the equation?
ANSWER: The following paragraph was incorporated in the manuscript:
Percent crystallinity was calculated using the following equation:
(Equation 1)
where is the melting enthalpy (Jg-1) of the polymer nanocomposite, is the enthalpy corresponding to the melting of a 100% crystalline sample (289 J g-1), and is the weight fraction of the filler in the nanocomposite (Wei, Tang, & Huang, 2004) . The standard deviation of the Tm and Tc measurements is ca. ±2 °C.
6. There are few grammar errors that have crept into the paper, for example: page 4, line 154: the FTIR spctrum...show", etc.
It was corrected by FTIR spectra
ANSWER: The new manuscript version was corrected by an experienced English speaker.
7. Did you estimate also Tc besides Tm. The analysis on the influence of the addition of nanofiller on the crystallinity of LDPE should be expanded upon the discussion on Tc. Maybe authors can add such data in the Table 1.
ANSWER: Yes, we estimated all transition temperatures and they were incorporated in Table 1. But the crystallization temperature did not show any change with the nanoparticle incorporation.
The crystallization temperature, melting temperature and crystallinity degree (cc) did not change with the nanoparticle incorporation, meaning that the presence of these nanoparticles did not affect the crystallization process of the polymer matrix.
8. In my opinion, not only T10 should be presented. Maybe also the beginning of degradation process - T2% and the middle of this process: T50% can be presented. Than one can observe how the addition of nanofillers affect the decomposition process on every step of degradation.
ANSWER: Table 1. was completed with T2% and T50%.
The new paragraph in the manuscript is:
In the initial degradation step of the decomposition temperature at 2% weight loss (T2), the nanocomposites (LDPE/CaCO3) were slightly more stable than LDPE nanocomposites with ca. 5%. For 10% weight loss (T10), for 50% weight loss (T50) and temperature for the maximum rate of weight loss (Tmax) did not change with the nanoparticle incorporation compared to the pure polymer under inert conditions.
9. Point 3.2.2-please provides the stress-strain curves for the samples. In my opinion the data presented in table are not enough for the analysis.
ANSWER: The measurement of the mechanical properties of the neat LDPE and the nanocomposites were repeated at least four times. Table 4, shows the new values and Figure 4 shows the stress-strain behavior.
10. There are only 8 from 29 references published from the late 5 years- which means less than 30% - maybe authors can find more actual references that can be cited in the paper.
ANSWER: New references were included in the manuscript, and references recommended by other reviews.
1. Kyzioł K., Oczkowska J., Kottfer D, Klich M., Kaczmarek L., Kyzioł A., Grzesik Z., Physicochemical and Biological Activity Analysis of Low-Density Polyethylene Substrate Modified by Multi-Layer Coatings Based on DLC Structures, Obtained Using RF CVD Method , Coatings ,2018, 8, 135
2. Lapcík L., Manas D., VasinaM., Lapcíkova B., Reznícek M., Zadrapa P, High density poly(ethylene)/CaCO3 hollow spheres composites for technical applications, Composites Part B 113 (2017) 218-224, http://dx.doi.org/10.1016/j.compositesb.2017.01.025).
3. Wong A.C-Y, Wong A.C.M.,, Extrudate swell ratio characteristics of CaCO3 added linear low density polyethylene, Polymer Testing 71 (2018) 262–271)
Reviewer 3 Report
I recommend the paper for publication in Molecules but after minor revision, please find the attached file.

Author Response
December 10, 2018 , Chile
Professor
Editor-in-Chief
Molecules
Dear Editor
First of all, we would like to thank the reviewers for their revision, which was very helpful to improve the manuscript.
Please find enclosed with this note our responses to the reviewers regarding their comments to our manuscript entitled: “Effect of CaCO3 nanoparticles on the mechanical and photo-degradation of LDPE ”
Reviewer 3
This paper describes research focused on selected mechanical and photo degradation properties of low density polyethylene (LDPE) substrate before and after modification of polymeric matrix using CaCO3 NPs (pure as well as oleic acid-modified). In summary, I recommend the paper for publication in Molecules after minor revision.
The study is noteworthy and extensive but it also presents some incompleteness, please see below:
1. Please provide in Introduction part more references concerning the research papers, based on promising (in recent years) surface plasma modification of LDPE substrate (incl.plasma technologies, surface functionalization and/or coating deposition) for increase physicochemical and mechanical as well as tribological (anti-wear) properties. These technologies are very useful and perspective for many application, e.g. in implanthology. It may be a paper such as:
Physicochemical and biological activity analysis of low-density polyethylene substrate modified by multi-layer coating based on DLC structures, obtained using RF CVD method, Coating, 8 (2018) 135
Influence of non-thermal plasma forming gases on improvement of surface properties of low density polyethylene (LDPE), Applied Surface Science. 307 (2014) 109-119.
ANSWER: The following paragraph was included in the new manuscript.
On the other hand, to improve the physicochemical properties of the polymer, some researchers have treated the surface polymer, using thin layer technology, including oxygen and nitrogen plasma discharge, deposition of functional coatings (i.e., diamond-like carbon (DLC) among others. For example, low density polyethylene increased its surface hardness 7 times after layer deposition by DLC coating. Those technics allowed giving desirable surface properties to the polymer.
2. Page 1, lines 24-25: Please add (in the last sentence of abstract) some information – more precisely specify the suggested field of application of modified polyethylene materials.
ANSWER: The new sentence incorporated into the manuscript is:
Our results show that the addition of CaCO3 nanoparticles into an LDPE polymer matrix allowed future developments of more sustainable polyethylene materials that could be applied as films in agriculture.
3. Page 1, line 39: “…and antimicrobial behavior [5].” Please confirm this statement, add suitable references.
ANSWER: Yes, It has been confirmed that t CaCO3 could have antimicrobial behavior. Other publications have been included to support this statement.
4. Page 2, lines 83-84: “The calcium nitrate solution was added drop wise to the sodium 83 carbonate solution with continous stirring during 4h.” At what temperature (RT)? Please add this information in the manuscript text.
ANSWER: The reaction temperature was 25 ºC, The reaction temperature was added into the manuscript.
5. Page 4, line 153: Please correct “Figure 2. XRD diffraction spectra of CaCO3 nanoparticles”. As “Figure 2. XRD spectra of CaCO3 nanoparticles.”
ANSWER: It was corrected
6. Page 5, line 160: Please correct “…a small signal at…” as “... a small spectra line at…”.
ANSWER: It was corrected in the new manuscript.
7. Page 6, lines 192-193 as well as Page 7, lines 230-231: Please calculate the measurement errors for all presented data in Table 1 and 3, respectively
ANSWER: The following sentence was included below Table 1.
The standard deviation of the viscosity measurements is ±0.03 dLg-1.The standard deviation of the Tm and Tc measurement are ca. ±2 °C. The TGA analysis has a standard deviation of ca. ± 2 ºC.
8. Page 6, Table 2 as well as Page 7, Table 4: Please correct “Mpa” as “MPa”. Do you measured the hardness modulus? If yes please add values of this parameter in tables.
ANSWER: Mpa was replaced by MPa. We did not measure the hardness modulus. We measured the Young’s modulus, which is reported in Table 2.
9. Page 8, line 266: Please correct “Figure 5. IR spectrum of…” as “Figure 5. IR spectra of…
ANSWER: It was corrected.
10. The conclusions are too general, and should be enriched by detailed information (parameters values or change in %) of obtained results.
ANSWER: The parameter values were changed into % and were incorporated into the manuscript. The new paragraph is:
Regarding to photoaging polymers, the degree of crystallinity (%c) , increased with photoaging and this effect was higher for LDPE/CaCO3 (ca. 19 %) nanocomposites than for neat LDPE (ca. 8%), attributed to recrystallization of the polymer. The viscosity of LDPE decreased by ca. 59% after photoaging and and around 72% for LDPE/CaCO3, as indicated by the decreased molecular weight of the polymer, due to chain scissions, and the pronounced effect of the nanoparticles in the polymer degradation. Young’s Modulus increased ca. 16 % for LDPE/O-CaCO3 after photoaging because the nanoparticles accelerate the polymer’s degradation.
Round 2
Reviewer 1 Report
See attachment

Author Response
December 18, 2018 , Chile
Professor
Editor-in-Chief
Molecules
Dear Editor
First of all, we would like to thank the reviewers for their revision, which was very helpful to improve the manuscript.
Please find enclosed with this note our responses to the reviewers regarding their comments to our manuscript entitled: “Effect of CaCO3 nanoparticles on the mechanical and photo-degradation properties of LDPE ”
Reviewer 1
Paper review for Molecules
Effect of CaCO3 nanoparticles on the mechanical and photo-degradation
properties of LDPE.
by Paula A. Zapata et al.
General comment
1. My original review of this paper was based on a complete misunderstanding of the purpose of it for which I apologize. I understood “improved photodegradation” as “less photodegradation” in order to make the material more resistant against environmental attack. But the contrary is the case! So why not replace “improved photodegradation” or “increased photodegradation” by “accelerated photodegradation”? This would make the point much more clear and prevent other readers from stepping into the same trap.
Answer: Thanks for the reviewer for the suggestion. In the new manuscript, we replaced in the statements “improved photodegradation” and “increased photodegradation” by “accelerated photodegradation”
2. Re-reading the paper in view of this new insight, I miss more hints to the environmental (pollution) aspect. I only found it in the last sentence of the abstract and in the paragraph line 301ff. Why not shifting the sentence line 301-302 to the beginning of the introduction? And/or even to the abstract?
Answer: Thanks to the Reviewer for the recommendation. We changed some sentence lines and the abstract according the suggestion. In particular, the abstract states the environmental issue in the beginning and the lines 301- 302 were moved to the beginning of the introduction.
Detail comments
3. New title
There is a “properties” missing after photo-degradation.
Answer: The new title is:
“Effect of CaCO3 nanoparticles on the mechanical and photo-degradation properties of LDPE”
4. Line 129ff . I would like to read here how thick the samples are (although it is mentioned in the previous paragraph) and whether the irradiation is from one side only or both (people using this irradiation apparatur will know, but I don’t)
Answer: The irradiation of the samples is from one side and it is alternated every 3 days. In the new manuscript the next sentence was added in the experimental part:
-Line 148 :Polymer films of 0.02 mm of thickness and the dimensions and 4 cm were irradiated
-Line 152: And the irradiation side of the sample is alternated every 3 days
5. Additional typos have been introduced with the new text: Lines 24,92.
Tables 2 and 4 , Shouldn’t E break read (epsilon) break?
Answer: It was corrected in the Table 2 and 4, you have reason is: e break.
Editor
In the revised version of your manuscript, please provide a sharper Figure 2 and for numbers ≥ ten thousand, please add a decimal comma in the Figure 2 (for example, you should change 10000" into "10,000").
Answer:
It was corrected in the new Figure 2.
Reviewer 2 Report
The paper has been improved following all of the reviewer's comments therefore I recoomend it for publication in the present form.
Author Response

(The authors gave the same response as above.)
